# Chronic and immediate refined carbohydrate consumption and facial attractiveness

**Amandine Visine, Valérie Durand, Léonard Guillou[¤], Michel Raymond[☉], Claire Berticat[☉]***

ISEM, Univ Montpellier, CNRS, EPHE, IRD, Montpellier, France

☉ These authors contributed equally to this work.
¤ Current address: Institut Jean Nicod, Département d'études Cognitives, Ecole Normale Supérieure, Universite PSL, EHESS, CNRS, Paris, France
* claire.berticat@umontpellier.fr

**Data Availability Statement:** The data and R script associated with this research are available on Zenodo repository 10.5281/zenodo.7708732 but the photos of the participants are not available.

## Abstract

The Western diet has undergone a massive switch since the second half of the 20th century, with the massive increase of the consumption of refined carbohydrate associated with many adverse health effects. The physiological mechanisms linked to this consumption, such as hyperglycaemia and hyperinsulinemia, may impact non medical traits such as facial attractiveness. To explore this issue, the relationship between facial attractiveness and immediate and chronic refined carbohydrate consumption estimated by glycemic load was studied for 104 French subjects. Facial attractiveness was assessed by opposite sex raters using pictures taken two hours after a controlled breakfast. Chronic consumption was assessed considering three high glycemic risk meals: breakfast, afternoon snacking and between-meal snacking. Immediate consumption of a high glycemic breakfast decreased facial attractiveness for men and women while controlling for several control variables, including energy intake. Chronic refined carbohydrate consumption had different effects on attractiveness depending on the meal and/or the sex. Chronic refined carbohydrate consumption, estimated by the glycemic load, during the three studied meals reduced attractiveness, while a high energy intake increased it. Nevertheless, the effect was reversed for men concerning the afternoon snack, for which a high energy intake reduced attractiveness and a high glycemic load increased it. These effects were maintained when potential confounders for facial attractiveness were controlled such as age, age departure from actual age, masculinity/femininity (perceived and measured), BMI, physical activity, parental home ownership, smoking, couple status, hormonal contraceptive use (for women), and facial hairiness (for men). Results were possibly mediated by an increase in age appearance for women and a decrease in perceived masculinity for men. The physiological differences between the three meals studied and the interpretation of the results from an adaptive/maladaptive point of view in relation to our new dietary environment are discussed.

**Funding:** This work was supported by Agence Nationale pour la Recherche "HUMANWAY" project (ANR-12- BSV7-0008-01). The funders had no role in study design, data collection and analysis, decision to publish, or preparation of the manuscript.

**Competing interests:** The authors have declared that no competing interests exist.

## Introduction

In Western populations, the diet has dramatically changed since the second half of the 20th century. It has been supplemented with highly processed refined food, in particular refined carbohydrates (primary sucrose, fiber depleted gelatinous starches and high sugar corn syrup [1]. The mismatch between how human physiology has evolved and Western industrialized lifestyles is seen as a contributing factor to the current epidemic of numerous medical problems. For example, it has been shown that this massive dietary change was involved in obesity, insulin resistance, type II diabetes, cardiovascular diseases, Alzheimer's disease, hypertension or myopia [2–5]. Persistent hyperglycemia/hyperinsulinemia, and insulin resistance due to the overconsumption of refined carbohydrates are among the well recognized physiological mechanisms involved in these diseases [1, 6, 7].

The consequences of a diet high in refined carbohydrates on non-medical traits have been little studied to date, and it is conceivable that they materialise in secondary sexual traits such as male or female facial characteristics [8]. Indeed, among other things, hyperinsulinemia modulates growth factors and sex hormones, interfering with morphology and secondary sex characteristics [9]. This occurs because hyperinsulinemia stimulates androgen synthesis by the ovaries or testes, increasing the quantity of free (and thus active) androgens in the blood which are the precursors of male and female sex hormones such as testosterone and estrogen [10–12]. As a consequence, for example, hyperinsulinemia has been linked to diseases associated with significant perturbation of sex hormone levels, such as polycystic ovary syndrome and premature menarche [13, 14]. Thus, considering that femininity and masculinity influence attractiveness [15], refined carbohydrate consumption via hyperinsulinemia could interact with attractiveness. Attractiveness is an important trait that affects a variety of key social outcomes such as mate choice and social exchange decisions. In the field of evolutionary biology, attractiveness (or preference) refers to an individual's tendency to be drawn to specific traits or characteristics in potential mating or social exchange partners. For example, people who are physically attractive (as opposed to unattractive) are more likely to be rated higher as romantic partners [16], students by teachers [17], and even as political candidates [18].

It has been recently proposed that refined carbohydrate consumption, particularly during between-meal snacks, might be correlated with an increase in facial attractiveness of both male and female individuals [8]. However, in this pilot study, subjects were sampled at various times of the day, thus at various times since their last meal, and the content of this last meal was not considered. However, immediate consumption of some food or drinks has detectable effects on physiology and behavior. For example, immediate alcohol consumption may influence facial attractiveness [19], and breakfast or snacks may have an immediate effect on behavioral components and cognition according to their glycemic load or energy intake [20, 21]. Additionally, several confounding variables were not controlled for, such as physical activity (which might indirectly influence attractiveness and diet). Also, energy intake was not considered in the statistical model. Thus, it is unclear from this preliminary study how refined carbohydrate consumption might influence attractiveness.

Different daily meals have different nutrient content and therefore do not produce the same glycemic response. In fact, carbohydrates are rarely eaten alone, and their digestion (degradation and absorption) can be affected by other macronutrients. Meals high in refined carbohydrates and low in fat, protein, and fiber result in a higher glycemic response [22, 23]. The sequence of dietary intake of macronutrients also alters glycemic and insulin responses [23]. Thus, meals described as high in refined carbohydrates and low in food content, such as breakfast, afternoon snacks, and between-meal snacks, may increase glycemic risk [20, 24, 25].

Here, we investigated whether refined carbohydrate intake affects facial attractiveness in healthy young women and men, accounting for several confounding variables and experimentally controlling for immediate carbohydrate intake. The present study is a replicate of this previous work [8] designed to overcome its weaknesses. Estimates of refined carbohydrate intake were based on the total glycemic load (representing blood glucose and insulin responses) of three meals with higher glycemic risk (breakfast, afternoon snack, and between-meal snack). The quantities consumed for each diet item were recorded, providing more relevant glycemic load estimation and both glycemic loads and energy intakes were independently considered, with the consequence that refined carbohydrate consumption better represented the effects of insulinemia. Subjects were given a high or low glycemic isocaloric breakfast, completed a dietary questionnaire, and were photographed at the same time after breakfast. This allowed us to examine how immediate and chronic consumption of refined carbohydrates affects facial attractiveness in healthy adults.

## Materials and methods

### Subjects

During 2018, subjects were invited to participate in a scientific study on diet via online calls, spread among to various university networks (Paul Valéry University, University of Montpellier, Engineering School Montpellier SupAgro) and social networks. The conditions for participation were being non diabetic and non hemophiliac, lacking food allergies, and without facial tattoo. Subjects were given an early-morning appointment and were asked to come to our laboratory for the experiments in groups of three or four on an empty stomach. They were given, at random, an isocaloric breakfast (approximately 500 Kcal) of type B1 (all carbohydrates were non refined) or B2 (all carbohydrates were refined) as described in [26].

The following data were self-reported for each subject: sex, age (birth year and month), sexual orientation (heterosexual or other), geographical origin of the grandparents (continents: Europe/Africa/ America/Asia/Oceania), couple status (yes or no, coded as 1 or 0, respectively), smoking (yes or no, coded as 1 or 0, respectively), parental home ownership as a proxy of socioeconomic status (owner or non-owner, coded as 1 or 0, respectively), physical activity (from 1: low activity, to 5: high activity), and, for women, use of hormonal contraceptive (yes or no, coded as 1 or 0, respectively). A quantitative diet questionnaire using the SU.VI.MAX cohort portion book concerning breakfast, afternoon snack ["goûter" in French, corresponding to an after-school snack] and between-meals snack of the day before was used to estimate participants' chronic refined carbohydrate consumption as previously described [26–28], S1 Table).

Then, the following body measurements were performed: height (using a measuring board) and weight (using a portable weighing scale). Subject glycemia was also measured 3 times (on an empty stomach, 30 minutes after breakfast and one hour and a half after breakfast), allowing the confirmation that consumption of refined carbohydrates during breakfast had a significant immediate effect on glucose metabolism [26].

Approximately two hours after they finished their breakfast, the subjects were photographed. Individual facial photographs were obtained from a frontal perspective at a distance of approximately 1 m using the same digital camera (Canon EOS 20D) with a 50-mm focal length in standardized settings (same room, light and white uniform background). The subjects were asked to express a neutral face (without a smile), to tie their hair and to remove any glasses, earrings, piercing and make-up. All photographs were processed using Adobe Photoshop CS5.1 to normalize size (photographs were aligned on the eye position, with a fixed distance between the eyes and the chin). An index of facial hairiness was estimated from male photographs, from 0 (no beard, no moustache) to 6 (abundant beard and moustache).

A total of 52 male and 52 female subjects with completely filled out questionnaires were finally selected according to the following characteristics: aged from 20 to 30, heterosexual, and with their 4 grand parents of European origin to reduce cultural heterogeneity. Descriptive statistics of physical characteristics of subjects are given in Table 1. The different food groups chronically consumed during these 3 meals are reported in Table 2. The proportion of individuals taking a breakfast was 87% for both sexes, those taking an afternoon snack were 38% and 52%, for males and females, respectively, and those taking a between-meal snack were 25% and 29%, respectively (Table 2).

## Daily diet variables

As previously described in [26], for each subject and each item of dietary questionnaire, Glycemic load (GL) was calculated by multiplying the glycemic index (GI) according to the International tables of glycemic Index [29] by the amount of available carbohydrates (g) per declared serving estimated from the SU.VI. MAX cohort catalogue divided by 100 [30]. Compared with low-GL diets, high-GL diets cause greater glycemic and insulin responses [29]. For each subject, the glycemic load for each item was then summed, leading to a total glycemic load (Table 3) estimation for breakfast (GL1), afternoon snack (GL2) and between-meal intake (GL3). In the same way, Energy intake (EI) for each item was obtained from the Anses-Ciqual database (www.anses.ciqual.fr) and calculated for each subject depending on its corresponding declared serving size (Table 3). For each subject, they were summed, leading to a total energy intake estimation for breakfast (EI1), afternoon snack (EI2) and between-meal intake (EI3) and corresponding macronutrient compositions. Mean glycemic load (GL) and energy intake (EI) for each meal were computed considering only consumers (Table 3). According to the general classification (e.g. [31]), the means of GL obtained for breakfast were high (>20).

To measure GL independent of EI, the method described in [32] was applied: linear models were used to produce regressions of EI as a function of GL for each meal using the lm function from the stats package for R software. These regression residuals were then used as new variables. Thus, each subject's refined residual carbohydrate consumption is now described for breakfast (RGL1), afternoon snack (RGL2) and between-meal intake (RGL3). These variables correspond to the part of the glycemic load that is not explained by energy intake.

## Femininity/Masculinity index

To generate a morphological facial femininity/masculinity index (Fem/Masc Index), a geometric morphometric analysis of the faces was performed following the methods described in [33–35]. First, the coordinates of 142 landmarks (anatomical points present in all individuals, e.g., lips corners) and semi landmarks (sliding points positioned along selected anatomical curves, such as the eyebrow bow) were delineated for each male and female face. Landmark and semi landmark delineation were performed using Psychomorph [36]. The R package Geomorph (version 4.0.0) was used to perform Procrustes superimposition of the landmark and semi landmark data, which removes non shape information such as translation, size and rotational effects [37]. The coordinates were transformed into shape variables via principal component analysis (PCA). An arbitrary cut-off of minimum 80% variance explained was applied to select the axis, thus the first 16 axes were retained (explaining 83.7% of variance) for further analyses. To compute a data-driven single measure of facial masculinity, a linear discriminant analysis (LDA) was conducted on the PCA coordinates with sex as the grouping variable. The resulting discriminant function correctly classified 90.4% of subjects into two categories (48 women and 46 men out of the 104 individuals; thus, 4 women and 6 men were not rightly sorted). Each

**Table 1. Descriptive statistics of the subjects' physical characteristics.**

|  | Women (N = 52) | | | Men (N = 52) | | |
|---|---|---|---|---|---|---|
|  | Range | Mean | SD | Range | Mean | SD |
| Age (years) | 20–28 | 22.5 | 2.0 | 20–30 | 23.0 | 2.1 |
| Perceived age (years) | 19–31 | 24.7 | 2.0 | 21–31 | 25.5 | 2.4 |
| BMI (kg/m$^2$) | 16–30 | 22.1 | 3.1 | 16–37 | 23.0 | 3.8 |
| Physical activity | 1–5 | 2.9 | 1.1 | 1–5 | 3.3 | 1.2 |
| Facial hairiness | - | - | - | 0–6 | 2.3 | 1.6 |
| Fem/Masc Index | -5.68 to -0.94 | -3.17 | 1.02 | -1.93 to 1.99 | 0.00 | 0.98 |
|  | Range | N | % | Range | N | % |
| Smoker | 0–1 | 2 | 0.04 | 0–1 | 5 | 9.6 |
| Parental home ownership | 0–1 | 45 | 86.5 | 0–1 | 47 | 90.4 |
| Couple status | 0–1 | 27 | 51.9 | 0–1 | 30 | 57.7 |
| Contraceptive | 0–1 | 34 | 65.4 | - | - | - |

individual coordinate on the woman-man axis was used as a Fem/Masc Index, with high values indicating a more masculine facial morphology [33–35].

## Apparent age

The apparent age of each subject was evaluated from their facial photographs using raters. Volunteer raters were recruited in public places in Montpellier, France. For each rater, sex, age (birth year and month), grand-parent geographical origin and study level were recorded.

An HTLM/PHP computer program was generated to present randomly drawn subject photographs to raters. Each rater estimated the age of 22 distinct subjects. Three photographs randomly chosen among those previously viewed were presented again at the end to estimate judgment reliability. Unreliable raters (with more than fifteen years for the sum of the absolute difference between real ages and attributed ages during the three judgments of reliability) or non-adult raters (less than 18 years old) were removed. If the rater took more than 60 s or less than 0.5 s for the response, the trial was removed. To reduce cultural heterogeneity, only raters with 4 grandparents of European origin were kept in the study. This led to a final sample of 77 raters (39 men and 38 women, age range: 18–56, mean age ± s.d.: 26.5 ± 8 years for men and 25.3 ± 10 years for women), resulting in a total of 820 age estimations towards women and 860 age estimations towards men and a mean of 16.5 (range: 11–22) estimations for each man and

**Table 2. Number of individuals consuming the different food groups for each meal.** N indicates the number of consumers.

|  | Women (N = 52) | | | Men (N = 52) | | |
|---|---|---|---|---|---|---|
| Food group | Breakfast (N = 45, 87%) | Afternoon snack (N = 27, 52%) | Between-meal snack (N = 15, 29%) | Breakfast (N = 45, 87%) | Afternoon snack (N = 20, 38%) | Between-meal snack (N = 13, 25%) |
| Cereals and bread | 34 | 9 | 2 | 35 | 6 | 3 |
| Biscuits, cakes, and pastries | 9 | 11 | 5 | 9 | 11 | 3 |
| Sweets and chocolate | 18 | 13 | 5 | 27 | 7 | 8 |
| Sweetened beverages | 15 | 5 | 0 | 14 | 3 | 3 |
| Dairy products | 31 | 5 | 3 | 22 | 8 | 3 |
| Fruits | 16 | 12 | 4 | 13 | 6 | 5 |
| Nuts | 5 | 5 | 2 | 2 | 0 | 0 |

**Table 3. Descriptive statistics of food consumption for the three meals.** Mean and standard deviation (SD) are given for consumers only. GL1, GL2 and GL3 are the three variables representing chronic refined carbohydrate consumption.

| | Women (N = 52) | | | Men (N = 52) | | |
|---|---|---|---|---|---|---|
| | Range | Mean | SD | Range | Mean | SD |
| Breakfast | | | | | | |
| GL1 | 0–92 | 26.6 | 23.0 | 0–142 | 38.5 | 33.3 |
| EI1 | 0–1039 | 334.1 | 265.0 | 0–1451 | 421.1 | 345.3 |
| Carbohydrates (g) | 0–144 | 37.5 (72%) | 33.7 | 0–224 | 53.2 (70%) | 44.9 |
| Fat (g) | 0–39 | 8.6 (16%) | 9.5 | 0–55 | 14.8 (20%) | 12.6 |
| Protein (g) | 0–27 | 6.0 (12%) | 5.8 | 0–36 | 7.8 (10%) | 8.0 |
| Fiber (g) | 0–17 | 3.3 | 3.5 | 0–22 | 4.2 | 4.3 |
| Afternoon Snack | | | | | | |
| GL2 | 0–62 | 11.8 | 16.5 | 0–118 | 12.6 | 23.4 |
| EI2 | 0–985 | 166.2 | 241.0 | 0–1065 | 148.2 | 264.5 |
| Carbohydrates (g) | 0–103 | 17.8 (64%) | 26.0 | 0–95 | 13.7 (74%) | 25.5 |
| Fat (g) | 0–72 | 7.1 (25%) | 13.3 | 0–42 | 2.9 (16%) | 7.0 |
| Protein (g) | 0–23 | 3.0 (11%) | 4.9 | 0–30 | 1.9 (10%) | 5.0 |
| Fiber (g) | 0–16 | 3.3 | 3.3 | 0–10 | 1.0 | 2.2 |
| Between-Meal Snack | | | | | | |
| GL3 | 0–42 | 3.7 | 8.9 | 0–110 | 6.9 | 18.4 |
| EI3 | 0–622 | 61.0 | 143.7 | 0–1010 | 78.5 | 192.0 |
| Carbohydrates (g) | 0–66 | 4.6 (59%) | 11.1 | 0–111 | 8.0 (73%) | 20.2 |
| Fat (g) | 0–39 | 2.2 (28%) | 6.4 | 0–23 | 1.8 (17%) | 5.2 |
| Protein (g) | 0–19 | 1.0 (13%) | 3.2 | 0–22 | 1.1 (10%) | 4.5 |
| Fiber (g) | 0–8 | 0.5 | 1.6 | 0–7 | 0.8 | 1.7 |

15.8 (range: 12–21) for each woman. The perceived age was on average 2.4 years older than the actual age (mean ± s.e.m. of 2.4 ± 0.21). It was estimated to be either younger (maximum 2.4 years) or older (maximum 6.2 years) than the actual age (Table 1).

## Perceived masculinity and femininity

The relative masculinity or femininity of each subject was assessed from the facial photographs using a second rater set. Volunteer raters were recruited in public places in Montpellier, France. For each rater, sex, age (birth year and month), sexual orientation, geographical origin of the grandparents and study level were recorded.

An HTLM/PHP computer program was generated to present randomly drawn pairs of same-sex photographs (Fig 1). Pairs were presented to opposite sex raters. For each male pair, the female raters were instructed to click on the photograph depicting the face that they found the most masculine. For each female pair, the men raters were instructed to click on the photograph they found the more feminine. The photograph position on the screen (left or right) was randomly ascribed. Each rater assessed 25 distinct pairs of photographs, corresponding to different randomly chosen subjects. Three pairs randomly chosen from among those previously viewed were presented again at the end to estimate judgment reliability. Unreliable raters (with more than one incorrect answer during the test of judgment reliability) or non-adult raters (less than 18 years old) were removed. If the rater took more than 60 s or less than 0.5 s for the response, the trial was removed. To reduce cultural heterogeneity, only heterosexual raters with 4 grand-parents of European origin were kept in the study. A total of 150 raters were retained in the final sample (68 men and 82 women, age range: 18–71, mean age ± s.d.: 35.5 ± 16 years for

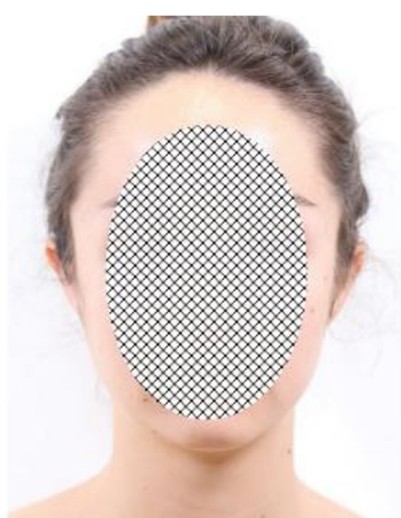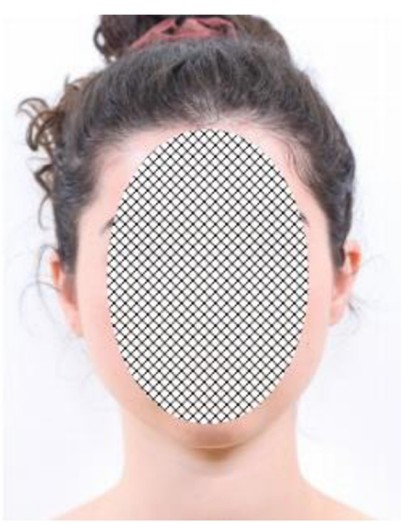

**Fig 1. Example of a pair of faces used during the evaluation of women's facial attractiveness by male raters.** For each pair of women, the rater was instructed to click on the photograph of the woman that he found the most attractive. Faces were anonymized for publication.

men, and 33.1 ± 14 for women), corresponding to 1,494 judgments of men towards women and 1,802 judgments of women towards men. Each subject was seen by a mean of 69.3 (range: 51–88) raters for men and a mean of 57.5 (range: 41–76) raters for women. No correlation for perceived masculinity was evidenced with the Fem/Masc Index for men (Pearson correlation coefficient = 0.20, p = 0.159). Perceived femininity was also not correlated (Pearson correlation coefficient = - 0.18, p = 0.196) with the Fem/Masc Index for women.

## Attractiveness

The subject relative attractiveness was assessed from their facial photographs using a third rater set. Volunteer raters were recruited in public places in Montpellier, France. For each rater, sex, age (birth year and month), sexual orientation, geographical origin of the grandparents and study level were recorded.

The same experimental protocol as for perceived masculinity and femininity was used to assess attractiveness. Raters assessed 26 distinct pairs of photographs, and three pairs randomly chosen from among those previously viewed were presented again at the end to estimate judgment reliability. Raters with more than one incorrect answer during the test of judgment reliability or non-adult raters (less than 18 years old) were removed. If the rater took more than 60 s or less than 0.5 s for the response, the trial was removed. To reduce cultural heterogeneity, only heterosexual raters with 4 grandparents of European origin were kept in the study. A total of 252 raters were retained in the final sample (110 men and 142 women, age range: 18–73, mean age ± s.d.: 35.3 ± 13 years for men and 35.6 ± 13 years for women), corresponding to 2,860 judgments of men towards women and 3,536 judgments of women towards men. Each

subject was seen by a mean of 136.0 (range: 107–164) raters for men and a mean of 110.0 raters (range: 83–137) for women.

## Statistical analyses

The following statistical analyses were performed using R software version 3.6.3 using the packages blme (v1.0–5, [38]), lme4 (v1.1–26), stats (3.6.3) and lavaan (v0.6–8, [39]). The variance inflation factor was computed using the vif.mer function adapted from the vif function of the R package rms (v6.2–0, [40, 41]). The effects of the rater's age and study level on their perception of subject's age, masculinity/femininity and attractiveness were tested before proceeding to the analyses of evaluators' preferences in terms of attractiveness, in order to take them into account in case of significance.

**Effects of rater characteristics on their perception of subject's age, masculinity/femininity and attractiveness.** *Age perception*. To understand the potential effects of rater characteristics on the age perception of subjects, a first model was used. The response variable was the estimated age, and the variables of interest were the rater characteristics (age, sex, and study level). The model also integrated the subject sex. Each subject was viewed by several raters, and each rater evaluated several subjects. Thus, linear mixed-effect models using the lmer function were used. Random slope effects on the raters and on the subjects were integrated into the models. The regression showed no significant effect ($p > 0.05$) of the rater characteristics on their perception of age (S2 Table). This allowed us to compute for each subject the mean perceived age as the average age estimated by all raters. *Masculinity/femininity perception*. A measure of perceived masculinity was computed for men as the number of times a given individual was chosen divided by the number of occurrences in the experiment. To assess a potential effect of rater age on masculinity perception, this measure was computed using only raters below or above the median age. The two resulting masculinity measures were not significantly different (Wilcoxon signed ranks test, $p = 0.96$, S3 Table). The measures were also computed from tercile ages, with no significant differences (Friedman two-way analysis of variance, $p = 0.10$). The same procedure was used to test the influence of rater study level on masculinity perception. Masculinity measures from raters below or above the median study level were not significantly different (Wilcoxon signed ranks test, $p > 0.94$) or among the three terciles (Friedman two-way analysis of variance, $p > 0.58$). A measure of femininity was computed for women in a similar way as for men. A potential effect of rater age or study level on femininity perception was evaluated as above, and no significant effect was found (age: for median, $p > 0.33$, for terciles, $p > 0.32$; study level: for median, $p > 0.29$, for terciles, $p > 0.94$, S3 Table). *Attractiveness perception*. A measure of perceived attractiveness was computed for women and men as the number of times a given individual was chosen divided by the number of occurrences in the experiment. To assess a potential effect of rater age on attractiveness perception, this measure was computed using only raters below or above the median age. For both sexes, the two resulting attractiveness measures were not significantly different (Wilcoxon signed ranks test, $p = 0.71$ and $p = 0.36$ for women and men faces, respectively, S4 Table). The measures were also computed from tercile ages, with no significant differences (Friedman two-way analysis of variance, $p = 0.63$ and $p = 0.48$ for women and men faces, respectively, S4 Table). The same procedure was used to test the influence of rater study level on attractiveness perception. For both sexes, attractiveness measures from raters below or above the median study level were not significantly different (Wilcoxon signed ranks test, $p = 0.67$ and $p = 0.94$ for women and men faces, respectively, S4 Table) or among the three terciles (Friedman two-way analysis of variance, $p = 0.86$ and $p = 0.87$ for women and men faces, respectively, S4 Table).

**Statistical analyses for perceived attractiveness.** A logistic regression was used to analyze the rater preferences. The binary response variable corresponded to being chosen or not for the focal subject (arbitrarily, the subject presented at the left position) during each pair presentation. Subjects and raters occurred repeatedly (each subject was viewed by several raters, and each rater evaluated several pairs of subjects) and were thus random-effects variables. Therefore, generalized linear mixed models with a binomial error structure were applied. To force the models to fit away from singularities, the Bayesian bglmer function was used. The maximum random-effects structure (intercept and slope) was tentatively included according to [42]. For each choice made by a rater, the difference (left minus right) between the focal RGL1 and the non-focal subject was calculated, and the same procedure was performed for RGL2, RGL3, EI1, EI2 and EI3. These differences were integrated into the model as the variables of interest. The difference between focal and non-focal subjects concerning their type of immediate breakfast was also integrated as a qualitative variable of interest with three modalities (B1 versus B2, same breakfast type and B2 versus B1). Because subject pairs were rated by the opposite sex (men rated by women and women rated by men), two models were performed, one for each subject's sex. For both, several control variables potentially affecting facial attractiveness were added: age, age departure from actual age (further referred to as 'age departure'), Fem/Masc index, perceived masculinity/femininity, BMI, physical activity, parental home ownership [43], smoking, couple status, hormonal contraceptive use (for women) and facial hairiness (for men). For each control variable, values associated with the left individual minus values associated with the right individual were computed. All quantitative variables were centered. The significance of each term was assessed from the model including all of the other variables. Because the dietary variables (RGL1, RGL2, RGL3, EI1, EI2, EI3 and breakfast type) could potentially affect certain control variables directly (e.g., age departure, Fem/Masc Index and perceived masculinity/femininity) or be affected by control variables (physical activity), this could indirectly influence the effect of the GL variables on the dependent variable. To evaluate this possibility, structural equation modeling was performed using the variables from the model displaying p < 0.1, conservatively. An attractiveness measure was constructed for each individual, computed as the number of times this individual was chosen over the number of occurrences. A hypothesized path model was constructed for each sex, incorporating linear regressions with the diet variables. For women, it aimed to explain attractiveness, age departure from actual age, perceived femininity and contraception with EI1, RGL1, RGL2, RGL3 and the breakfast type. Parental home ownership, physical activity and age were also included as control variables (Fig 2). For men, the model incorporated linear regressions of EI1, EI2, RGL2, RGL3 and the breakfast type to explain attractiveness, Fem/Masc Index and perceived masculinity (Fig 3). Facial hairiness, physical activity, age and couple status were also included as control variables.

**Assessment of systematic and random errors.** We have sought to reduce the influence of systematic and random errors in our study and ensure the robustness of our results. To assess systematic errors, we meticulously designed our study by following established protocols and ensuring that experimental conditions were standardized as much as possible for participants and raters. This included the fact that we implemented carefully recruitment procedures to limit the risk of selection bias by defining clear inclusion and exclusion criteria for our study samples. In addition, all assessments of photographs and facial attractiveness were carried out in a controlled environment, with constant lighting, background and ambient temperature, to reduce variability caused by external factors. We used validated questionnaires. Participants and raters were not informed of the study objectives in order to minimize potential biases related to knowledge of the research objectives. We used appropriate statistical methods to control for potential confounding factors concerning subjects and raters (see Statistical

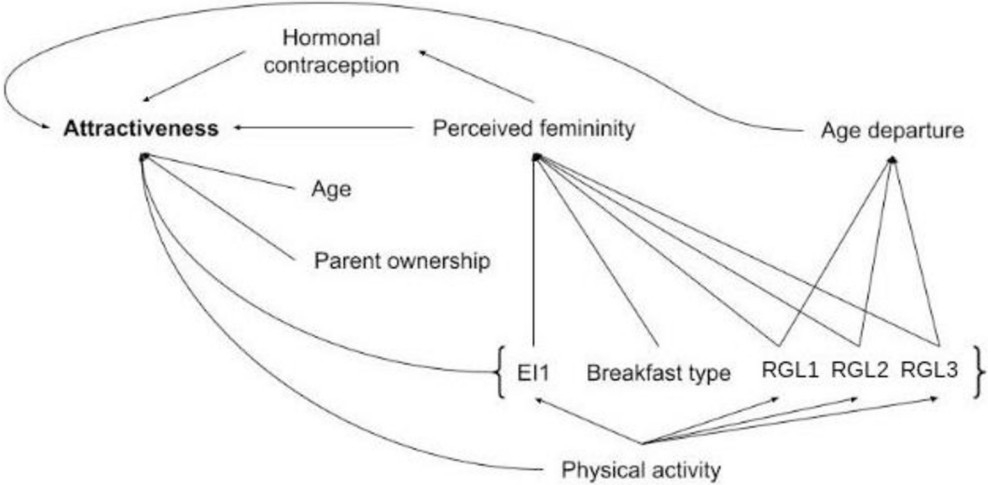

**Fig 2. Hypothesized path model for women's attractiveness with the variables of the generalized linear mixed model.**

Analyses section). To minimize random errors and improve the reliability of our measurements, we used highly reliable and reproducible measurement techniques for the photographs and for the assessment of facial attractiveness. The evaluation process was conducted randomly and independently by several raters to ensure the robustness of our results. The sample sizes of participants and raters were determined on the basis of an effect size of 0.1 (based on the results of a previous pilot study [8]), a power of 80% and a threshold of 0.05 to ensure that the study had sufficient statistical power to detect significant differences in attractiveness ratings, and were chosen to minimize the impact of random variability.

## Ethical statement

The protocol used to recruit participants and collect data was approved by the French Committee of Information and Liberty (CNIL #1783997V0) and the Committee for the protection

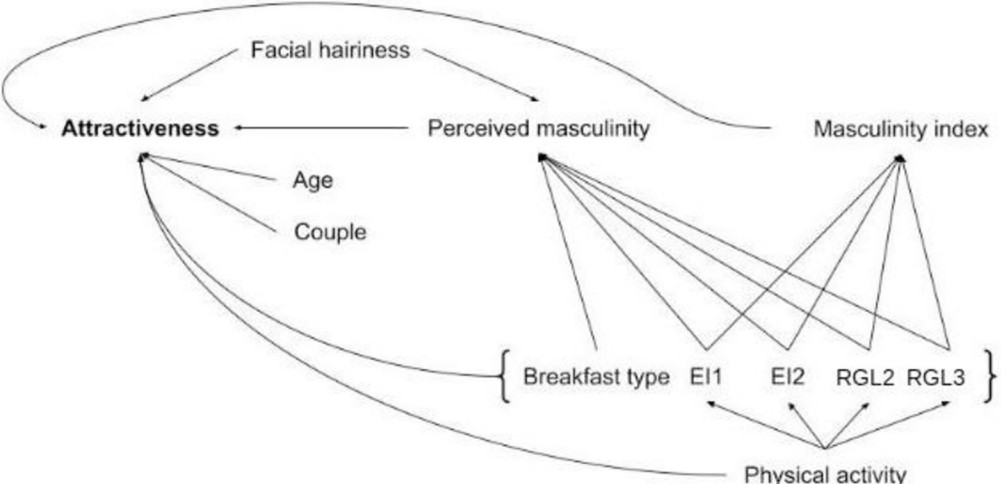

**Fig 3. Hypothesized path model for men's attractiveness with the variables of the generalized linear mixed model.**

of persons (CPP IDRCB 2018-A00505-50). For each participant, the general purpose of the study was explained ("Effects of diet on major phenotypic traits"), and written voluntary agreement was requested for statistical use of data (private information and photographs). Data were analyzed anonymously and no authors had access to information that could identify individual participants during or after data collection.

## Results

Chronic and immediate refined carbohydrate consumption, energy intake and controlling variables had different effects on attractiveness (Fig 4 and Table 4). The breakfast consumed by subjects just before the photo session had a significant effect on attractiveness for both sexes. Individuals who had B2 were considered less attractive than those with B1 (men: $\beta$ = -1.01, se = 0.187, p < $10^{-6}$; women: $\beta$ = -1.31, se = 0.191, p < $10^{-10}$). Some chronic diet variables of interest had the same effect between men and women: the probability that a subject was chosen as the most attractive was significantly influenced by the energy intake variable EI1 (men: $\beta$ = 0.26 se = 0.057, p < $10^{-5}$; women: $\beta$ = 0.38, se = 0.097, p < $10^{-3}$). For breakfast, women preferred men, and men preferred women, with the highest energy intake. The variable RGL3 decreased attractiveness; this effect was marginally non-significant for women ($\beta$ = -0.274, se = 0.158, p = 0.083) and significant for men ($\beta$ = -0.434, se = 0.089, p < $10^{-5}$). For RGL1, RGL2 and EI2, preference was influenced by different meals and had different directions for men and women: RGL1 and RGL2 significantly decreased women's attractiveness (RGL1: $\beta$ = -0.202, se = 0.097, p = 0.037; RGL2: $\beta$ = -0.235, se = 0.098, p = 0.017), whereas for men, RGL2 increased attractiveness ($\beta$ = 0.410, se = 0.082, p < $10^{-6}$) and EI2 decreased attractiveness ($\beta$ = -0.315, se = 0.079, p < $10^{-4}$). Men preferred women with lower breakfast and afternoon snack glycemic load, and women preferred men with a higher afternoon snack glycemic load and a lower energy intake.

Some control variables significantly influenced the choice of raters. For male and female subjects, age decreased the probability of being chosen as the most attractive (men: p < $10^{-4}$; women: p < $10^{-4}$). The age departure from actual age decreased women's attractiveness (p < $10^{-4}$): at equal actual age, men preferred women with the youngest perceived age. As expected, physical activity had an increasing effect for both sexes but effect was marginally non-significant for men (men: p = 0.063; women: p < $10^{-3}$): individuals with higher physical activity were found to be more attractive. Perceived masculinity/femininity significantly increased attractiveness (men: p = 0.042; women: p = 0.004), while the effect of Fem/Masc Index on increasing male attractiveness was marginally non-significant (p = 0.062). Women preferred men considered more masculine, and men preferred women considered more feminine. Facial hairiness decreased the probability of being chosen (p < $10^{-5}$): men with more abundant facial hairiness were considered less attractive by women. Women also preferred men who were involved in a couple (p = 0.002). Parental home ownership increased women's attractiveness (p = 0.007), and the use of a hormonal contraceptive also increased women's attractiveness (p < $10^{-5}$).

The full models for men and women explained 4.5% and 6.0% of the total deviance, respectively. For both, the variance inflation factors (VIFs) were less than 2.50 (less than 3.20 for the qualitative variable of breakfast type for men). The VIF values for both models indicated that the multicollinearity between covariables was weak and not of concern [41].

Structural equation modeling indicated that the effect of RGL2 for women on attractiveness could be mediated by an effect on age departure from actual age (p = 0.013, S5 Table): a higher snack glycemic load increased the appearance of women towards an older age. These analyses also showed that the effect of physical activity on women's attractiveness could be mediated by

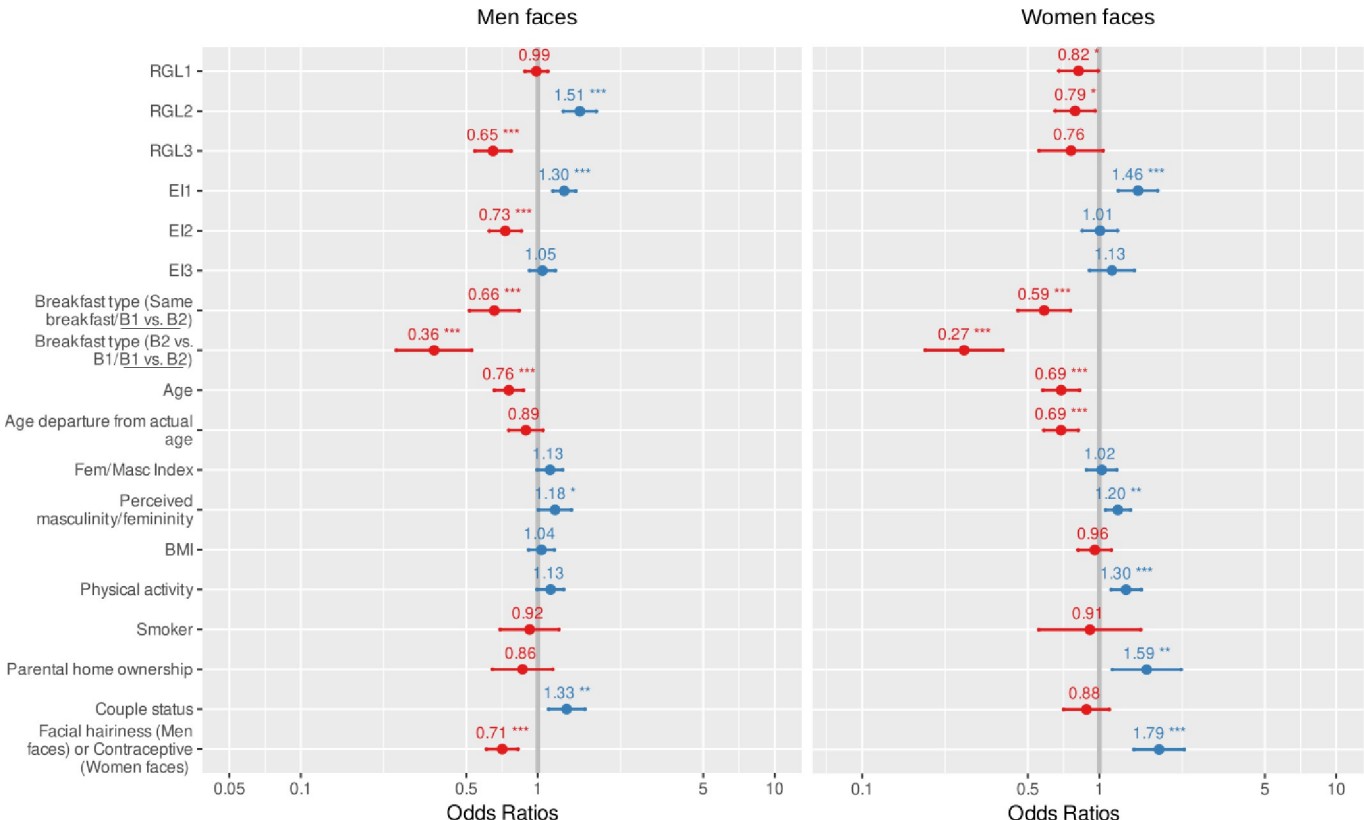

**Fig 4. Graphical representation of the adjusted odd ratios with their 95% confidence intervals from the model studying the probability of being chosen in the attractiveness test for male or female faces.** Raters were instructed to choose the individual found to be the most attractive between two facial photographs. RGL1, RGL2 and RGL3 are the three variables representing chronic refined carbohydrate consumption. For each variable, the difference between the two individuals (left minus right) presented was integrated into the model. For the immediate breakfast type variable, estimates are given for one category compared with the reference category corresponding to focal with B1 and non-focal with B2 (underlined term). * p < 0.05 ** p < 0.01 *** p < 0.001.

an effect on EI1, although marginally non-significant (p = 0.08): women who had higher physical activity were more likely to have breakfast with higher energy intake. For men, structural equation modeling indicated that the effect associated with RGL3 and EI2 on attractiveness could be mediated by an effect on perceived masculinity (RGL3: p = 0.015; EI2: p = 0.01, S5 Table). Men having a higher between-meal glycemic load or a higher energy intake during snacks were perceived as more feminine by women. The effect on EI1 on male attractiveness could also be mediated by an effect on perceived masculinity but in an opposite direction, although marginally non-significant (p = 0.057). Men with a higher breakfast energy intake were perceived as more masculine by women. Concerning the effect of physical activity on male attractiveness, analyses suggested that it could be mediated by RGL3 (p = 0.0140). Men practicing more sports had a lower between-meal glycemic load.

## Discussion

In this study, we investigated the relationship between refined carbohydrate intake and facial attractiveness in healthy adults, women and men. We observed that facial attractiveness is not independent of immediate or chronic consumption of refined carbohydrates. Immediate consumption of a high glycemic breakfast decreases facial attractiveness for men and women. Chronic refined carbohydrate consumption displays different effects on attractiveness

**Table 4. Effects of different variables on the probability of being chosen during the test of attractiveness for male or female faces.** Raters were instructed to choose the individual found to be the most attractive between two facial photographs. RGL1, RGL2 and RGL3 are the three variables representing chronic refined carbohydrate consumption. For each variable, the difference between the two individuals (left minus right) presented was integrated into the model. For the immediate breakfast type variable, the estimates are given for one category compared with the reference category corresponding to focal with B1 and non-focal with B2 (underlined term). The estimate (b), standard error of the mean (se), $\chi^2$ statistic, and corresponding p-value are given. F Bold characters indicate significant (p < 0.05) effects.

| | Male faces evaluated by women | | | | Female faces evaluated by men | | | |
|---|---|---|---|---|---|---|---|---|
| | β | se | $\chi^2$ | p($>\chi^2$) | β | se | $\chi^2$ | p($>\chi^2$) |
| Intercept | 0.455 | 0.160 | | | 0.558 | 0.180 | | |
| RGL1 | - 0.014 | 0.056 | 0.063 | 0.802 | - 0.202 | 0.097 | 4.349 | **0.037** |
| RGL2 | 0.410 | 0.082 | 25.23 | $< 10^{-6}$ | - 0.235 | 0.098 | 5.673 | **0.017** |
| RGL3 | - 0.434 | 0.089 | 23.64 | $< 10^{-5}$ | - 0.275 | 0.158 | 3.008 | 0.083 |
| EI1 | 0.259 | 0.057 | 20.30 | $< 10^{-5}$ | 0.377 | 0.097 | 15.05 | $< 10^{-3}$ |
| EI2 | - 0.315 | 0.079 | 15.88 | $< 10^{-4}$ | 0.006 | 0.088 | 0.005 | 0.942 |
| EI3 | 0.045 | 0.064 | 0.500 | 0.480 | 0.124 | 0.112 | 1.222 | 0.269 |
| Breakfast type (Same breakfast/B1 vs. B2) | - 0.421 | 0.123 | 30.70 | $< 10^{-6}$ | - 0.534 | 0.129 | 48.41 | $< 10^{-10}$ |
| (B2 vs. B1/B1 vs. B2) | - 1.008 | 0.187 | | | - 1.313 | 0.192 | | |
| Age | - 0.281 | 0.072 | 15.24 | $< 10^{-4}$ | - 0.370 | 0.091 | 16.65 | $< 10^{-4}$ |
| Age departure from actual age | - 0.116 | 0.094 | 1.895 | 0.169 | - 0.371 | 0.085 | 18.99 | $< 10^{-4}$ |
| Fem/Masc Index | 0.119 | 0.063 | 3.490 | 0.062 | 0.023 | 0.075 | 0.094 | 0.759 |
| Perceived masculinity/femininity | 0.167 | 0.085 | 4.133 | **0.042** | 0.182 | 0.062 | 8.460 | **0.004** |
| BMI | 0.035 | 0.064 | 0.311 | 0.577 | - 0.044 | 0.083 | 0.284 | 0.594 |
| Physical activity | 0.124 | 0.066 | 3.458 | 0.063 | 0.261 | 0.075 | 12.12 | $< 10^{-3}$ |
| Smoker | - 0.079 | 0.146 | 0.295 | 0.587 | - 0.092 | 0.252 | 0.132 | 0.716 |
| Parental home ownership | - 0.148 | 0.149 | 0.983 | 0.321 | 0.461 | 0.170 | 7.315 | **0.007** |
| Contraceptive | - | - | - | - | 0.581 | 0.124 | 21.95 | $< 10^{-5}$ |
| Couple status | 0.282 | 0.090 | 9.765 | **0.002** | - 0.125 | 0.113 | 1.237 | 0.266 |
| Facial hairiness | - 0.345 | 0.078 | 19.73 | $< 10^{-5}$ | - | - | - | - |

depending on the meal and/or the sex. Chronic refined carbohydrate consumption, estimated by the glycemic load, during the three studied meals (breakfast, afternoon snack and between-meal intake) reduced attractiveness, while a high energy intake increased it. Nevertheless, the effect was reversed for men concerning the afternoon snack, for which a high energy intake reduced attractiveness and a high glycemic load increased it. These effects were maintained when potential confounders for facial attractiveness were controlled such as age, age departure from actual age, masculinity/femininity (perceived and measured), BMI, physical activity, parental home ownership, smoking, couple status, hormonal contraceptive use (for women), and facial hairiness (for men).

## How refined carbohydrate consumption could affect facial attractiveness?

Immediate breakfast consumption influenced attractiveness. Women and men who had eaten a high-glycemic breakfast were considered less attractive than those who had eaten a low-glycemic breakfast. The two types of breakfast were isocaloric, although they differed in the resulting glycemic dynamics [28]. Two hours after breakfast consumption, when facial pictures were taken, only the high-glycemic breakfast generated hypoglycemia [44]. Hypoglycemia is known to have visible symptoms, as it affects blood flow and skin [45–47], which could be detectable on photos and thus affect attractiveness perception.

Chronic refined carbohydrate-rich food consumption leads to chronic hyperinsulinemia as a consequence of hyperglycemia, which interferes with growth factors and sex hormones, which in turn could modulate morphology and secondary sex characteristics [9]. Moreover,

saturated fat is a known antagonist of insulin and a contributor to insulin resistance [48]. Thus, a large energy intake due to saturated fat consumption, even associated with low refined carbohydrate consumption, could lend some support to the hyperinsulinemic theory of [9, 12]. Chronic hyperinsulinemia influences the synthesis of androgens which are the precursors of male and female sex hormones [49]. It has been shown that facial femininity/masculinity can be influenced by sex hormones, which in turn could affect attractiveness: in general men prefer more feminine faces and women prefer more or less masculine faces, depending on the tradeoff between the costs and benefits of mating with a masculine male (for a review see [15]). In this study, for men, the perceived masculinity and the morphological Fem/Masc indices were positively linked with attractiveness. Structural equation modeling suggested that the negative effect of chronic between-meal glycemic load and afternoon snack energy intake on attractiveness could be mediated by an effect on perceived masculinity (chronic glycemic load reducing masculinity, thus indirectly decreasing attractiveness).

Repeated hyperglycemia due of chronic consumption of refined foods rich in carbohydrates could also have an impact on facial attractiveness. Indeed, it has been shown that chronic hyperglycemia accelerates glycation processes which, in turn, have an impact on skin aging [50, 51]. As skin aging directly impacts age appearance [52], hyperglycemia could affect age perception. Moreover, age is known to influence attractiveness [53]. For women, this influence is generally negative, as men generally prefer younger women [53]. For men, preference studies based only on facial photographs (thus no information on social status) have found a decrease in men's attractiveness with age [54–56]. For women, both actual age and age departure from actual age lead to a decrease in attractiveness. Structural equation modeling showed that the effect of afternoon snack glycemic load on attractiveness could be indirectly mediated through a direct effect on age departure from actual age, leading to an older appearance at equal actual age.

## Why could the three chronic meals affect facial attractiveness differently between sexes?

Lipid and glucose metabolism are tuned to distinct sex-specific functions under the action of sex chromosomes and hormones. Considering glucose metabolism, women have higher whole-body insulin sensitivity than men [57, 58]. Glucose homeostasis, prediabetic syndromes, and type 1 and 2 diabetes show strong sex differences with a partial role of sex hormones [59]. The prevalence of type 2 diabetes and insulin resistance is higher in men than in women, and the opposite pattern is found for obesity [58]. Obese men are characterized by a progressive decrease in testosterone levels with increasing body weight, whereas obese women, particularly those with the abdominal phenotype (i.e., with insulin resistance), tend to develop a condition of functional hyperandrogenism [60]. Thus, the consequences of hyperglycemia and hyperinsulinemia could be different between men and women. For women, because of their higher sensitivity to insulin than men (and thus their lower risk of developing insulin resistance), sex hormones, and consequently facial femininity/masculinity, could be less affected by a large consumption of refined carbohydrates. This could explain why the effects of refined carbohydrate consumption on attractiveness were mediated by perceived masculinity for men, but not by perceived femininity for women.

## Why do the three chronic meals affect facial attractiveness differently?

For women, chronic refined carbohydrate consumption during breakfast decreased facial attractiveness, whereas energy intake increased it. The effect of energy intake was estimated in the model at equal glycemic load and thus primarily represented the effects of fat and protein

intake. Thus, breakfasts resulting in an increase in attractiveness comprised mainly fats and proteins (such as dairy) with few refined carbohydrates (Table 2). Breakfast is an important meal of the day, and skipping it (for those who usually take one) may be related to health issues, such as overweight and obesity [61–63] or bad health habits. For instance, skipping breakfast is linked to a decrease in physical activity in women [64]. Structural equation modeling showed a marginally non-significant effect of physical activity on women's breakfast energy intake. Indeed, women exercising more could be likely to have a higher protein and fat and a lower refined carbohydrate intake at breakfast, and to be involved in better life hygiene with a higher diet quality. Moreover, the intensity of physical activity can be a strong indicator of attractiveness [65], principally because it shows good health and because good health is associated with facial cues that affect attractiveness perception [66]. For men, an increase in energy intake during breakfast also increased attractiveness, probably for the same physiological and environmental reasons as those for women. However, the reduced attractiveness resulting from an increase in glycemic load during breakfast was restricted to women.

Afternoon snacking, a usual mid-afternoon meal known called "goûter" in France, corresponds (for people who are used to it) to a real food need. This meal is associated with a pre-prandial drop in plasma glucose and insulin concentrations and a strong motivation to eat [67], although only men were evaluated in that study. This could explain the increased attractiveness of men with a high glycemic load food consumption during the afternoon snack, providing immediately available glucose. Interestingly, also for men, energy intake had a reversed effect during this meal. One possibility is that a large proportion of saturated fat is involved in this meal, such as those found in pastries, as such fat is known to be an antagonist of insulin and a contributor to insulin resistance and thus hyperinsulinemia [48], thus mimicking the potential negative effects of refined carbohydrates on masculinity/femininity. This hypothesis was supported by structural equation modeling, which suggested that the decrease in attractiveness due to afternoon snack energy intake was mediated by a decrease in perceived masculinity for men. For women, the results were different: a negative effect of afternoon snack glycemic load on attractiveness was observed, and this effect was mediated by an older appearance (at equal chronological age), probably due to the consequence of hyperglycemia on aging.

Between-meal snacks are generally not associated with physiological hunger and are instead a consequence of social or other external stimuli, with little impact on satiety and compensation mechanisms [20]. The decreased attractiveness associated with an increase in refined carbohydrates consumption during between-meal snacks was observed for both sexes. For men, it could be modulated by physical activity and mediated by masculinity, as suggested by structural equation modeling: men who exercise less tend to eat more refined carbohydrates outside of regular meals, affecting their masculinity and their attractiveness.

Thus, the three types of meals might affect subjects' facial attractiveness differently because they correspond to different ecological eating habits that have different physiological consequences.

## How can the influence of diet on facial attractiveness be evolutionary triggered?

In general, traditional foods (pre-industrial or non-refined) do not generate hyperglycemia, with the exception of ripe fruits or honey which are energetically rewarding but are traditionally seasonal or scarce. In fact, humans did not evolve with constant access to food provoking a high glycaemic response, even after the rise of agriculture in the Neolithic era. It has been previously proposed that in the current industrial dietary environment, consumption of food that generates hyperglycemia is no longer a signal of quality, because this type of food is now not

limited [8]. Its massive consumption generates phenotypic and physiological changes in the body, such as obesity and type 2 diabetes, which are attracting medical attention due to their life-threatening effects. It is thus not surprising that other negative effects not directly affecting health are also generated, such as reduced facial attractiveness.

## Limitations

In this study, the chronic effect of refined carbohydrate intake on attractiveness may have been confounded by several variables that were not considered here. For example, attractiveness was not controlled for skin color (redness, yellowness) and aspect (brightness, luminance), although these factors are known to impact health perception [68] and could thus impact attractiveness [69, 70]. However, all facial photographs were taken indoors in the same technical room with fully controlled lighting conditions, thus reducing environmental variance for these traits. Skin color can also be modulated by diet and health habits [71–74]. For instance, fruit and vegetable consumption is known to increase skin yellowness [75, 76]. In addition, because lunch and dinner were not recorded in this study, it was not possible to calculate an overall index of diet quality that could have accounted for other aspects of food that influence attractiveness. However, diet quality index and glycemic load values are correlated: higher index values are associated with increased low GL foods, see, e.g., [77, 78] and high fruit and vegetable consumption. Thus, diet quality with fruit and vegetable consumption was partially described by glycemic load measures. Another potential confounding variable we do not control is menstrual cycle, in woman sample of subject and raters. It has been shown that facial attractiveness may increase during ovulation, as assessed by male raters [79] and that women's perception of men's facial attractiveness could be influenced by their menstrual cycle [80]. We only control for contraception in woman sample of subjects (65% of the sample took a contraception). However, the bias associated with not taking this variable into account could be offset by the sample size, with the assumption that each participant's menstrual cycle stage is randomized for the day of sampling. In addition, future studies should aim to control sleep, as sleep deprivation has been shown to have an effect on facial attractiveness [81]. Finally, the sample size of participants was relatively small. However, when 10% of the dataset was randomly deleted (1000 repetitions), for both women and men, the results did not change qualitatively (S6 Table). This indicates that the effects observed are strong enough to be detected even in a smaller sample.

## Conclusion

The recent Western dietary change, mainly the massive increase in refined carbohydrate consumption, has well described adverse health consequences. Traits not under medical competence but still with large social importance seem also impacted, such as facial attractiveness. Facial attractiveness, an important factor of social interactions, seems to be impacted by immediate and chronic refined carbohydrate consumption. Further studies are needed to investigate how diet effects are mediated and which other social traits could be affected by refined carbohydrate consumption.

## Supporting information

**S1 Table. Exhaustive list of the different food items of the diet questionnaire in French and translated.**
(DOCX)

**S2 Table. Effects of rater characteristics and subject age and sex on the subject age perception by raters.** Raters were instructed to ascribe an age for the photographs they were viewing. The estimate (β), standard error of the mean (se), $\chi^2$ statistic, and corresponding p-value are given. Bold characters indicate significant ($p < 0.05$) effects.
(DOCX)

**S3 Table. Effects of rater characteristics on the subjects' masculinity/femininity perception by raters.** The Wilcoxon test statistic (V), Friedman chi-squared (F) and corresponding p-value are given. Bold characters indicate significant ($p < 0.05$) effects. Median and terciles of age and study level were used for the Wilcoxon signed-rank test and Friedman two-way analysis of variance, respectively.
(DOCX)

**S4 Table. Effects of rater characteristics on the subjects' attractiveness perception by raters.** The Wilcoxon test statistic (V), Friedman chi-squared (F) and corresponding p-value are given. Bold characters indicate significant ($p < 0.05$) effects. Median and terciles of age and study level were used for the Wilcoxon signed-rank test and Friedman two-way analysis of variance, respectively.
(DOCX)

**S5 Table. Structural equation analysis.** The results based on Figs 2 and 3. RC1, RC2 and RC3 are the three variables representing refined carbohydrate consumption. The standardized estimate (β), standard error of the mean (se), z-value, and corresponding p-value are given. Bold characters indicate significant ($p < 0.05$) effects. Foxing each sex, only variables with a p-value $< 0.01$ were integrated into the model.
(DOCX)

**S6 Table. Sensitivity analysis for the test of attractiveness for male or female faces.** After a random 10% data reduction, p-values are computed and this process is repeated 1000 times, providing a p-value distribution for each variable. RGL1, RGL2 and RGL3 are the three variables representing refined carbohydrate consumption. The mean p-value (mean p), standard deviation (sd), minimum p-value (min) and maximum p-value (max) are given.
(DOCX)

## Acknowledgments

The authors thank the City Hall of Montpellier, Luc Gomel and all staff from Serre Amazonienne for providing places for rater recruitment and the women and men who participated in this study. This is contribution ISEM 2024–032.

## Author Contributions

**Conceptualization:** Michel Raymond, Claire Berticat.

**Data curation:** Léonard Guillou, Claire Berticat.

**Formal analysis:** Amandine Visine, Léonard Guillou, Claire Berticat.

**Investigation:** Amandine Visine, Valérie Durand, Léonard Guillou, Claire Berticat.

**Methodology:** Léonard Guillou, Claire Berticat.

**Supervision:** Michel Raymond, Claire Berticat.

**Validation:** Michel Raymond, Claire Berticat.

**Writing – original draft:** Amandine Visine, Michel Raymond, Claire Berticat.

**Writing – review & editing:** Michel Raymond, Claire Berticat.

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
