## [Decision Letter · Decision Letter 0]

5 Sep 2023

PONE-D-23-07591Chronic and immediate refined carbohydrate consumption measured by glycemic load, and facial attractivenessPLOS ONE

Dear Dr. Berticat,

Thank you for submitting your manuscript to PLOS ONE. After careful consideration, we feel that it has merit but does not fully meet PLOS ONE’s publication criteria as it currently stands. Therefore, we invite you to submit a revised version of the manuscript that addresses the points raised during the review process.

We look forward to receiving your revised manuscript.

Kind regards,

Shen Liu, Ph.D

Academic Editor

PLOS ONE

2. Please amend either the abstract on the online submission form (via Edit Submission) or the abstract in the manuscript so that they are identical.

3. We note that Figure 1  in your submission contain copyrighted images. All PLOS content is published under the Creative Commons Attribution License (CC BY 4.0), which means that the manuscript, images, and Supporting Information files will be freely available online, and any third party is permitted to access, download, copy, distribute, and use these materials in any way, even commercially, with proper attribution. For more information, see our copyright guidelines: http://journals.plos.org/plosone/s/licenses-and-copyright. 

A. You may seek permission from the original copyright holder of Figure 1 to publish the content specifically under the CC BY 4.0 license. 

B. If you are unable to obtain permission from the original copyright holder to publish these figures under the CC BY 4.0 license or if the copyright holder’s requirements are incompatible with the CC BY 4.0 license, please either i) remove the figure or ii) supply a replacement figure that complies with the CC BY 4.0 license. Please check copyright information on all replacement figures and update the figure caption with source information. If applicable, please specify in the figure caption text when a figure is similar but not identical to the original image and is therefore for illustrative purposes only.

Reviewers' comments:

Reviewer's Responses to Questions

**Comments to the Author**

1. Is the manuscript technically sound, and do the data support the conclusions?

Reviewer #1: Yes

Reviewer #2: Partly

2. Has the statistical analysis been performed appropriately and rigorously? 

Reviewer #1: Yes

Reviewer #2: Yes

3. Have the authors made all data underlying the findings in their manuscript fully available?

Reviewer #1: Yes

Reviewer #2: No

4. Is the manuscript presented in an intelligible fashion and written in standard English?

Reviewer #1: Yes

Reviewer #2: Yes

5. Review Comments to the Author

Reviewer #1: I read the whole manuscript entitled " Chronic and immediate refined carbohydrate consumption measured by glycemic load, and facial attractiveness" and I have few suggestions.

It's very hard to state attractiveness in words but it would be great if authors can define in few words/lines about it. Also, the opposite genders always have some degree of attractiveness to each other too. So, a formal statement about it should also be included in definition.

Page#3 and Line#64: I suggest changing the word “repeated” with persistent.

Page#3 and Line#68-69: any reference to support this would be great

Reviewer #2: Thank you for the opportunity to review this very interesting study.

This is an observational study assessing the association between refined carbohydrate consumption and facial attractiveness. The authors have conducted a thorough investigation, taking into account a large number of variables that could potentially be related to the main outcome.

Although I find great interest in the study and its results, I have some concerns regarding the methodology and study design. I find that the authors have not controlled well all possible confounders and have conducted a large number of tests in order to explore various associations. Also, the sample population is very small in order to make reliable conclusions about the primary outcome if all possible confounders are taken into account.

Therefore, I am unfortunately not able to recommend publication of this manuscript in its current form. My detailed comments are listed below.

Major comments:

•Introduction

- I suggest that the authors rephrase and soften their statements about the association between carbohydrate consumption and medical disease. Despite current evidence regarding these relationships, most of these diseases have multifaceted etiologies and are likely not attributed to a single cause. I am referring to the first paragraph of the introduction.

- The authors are citing Reference #8 to support a lot of their statement in the introduction. Reference #8 is a non-systematic, topic review article with little scientific merit. I suggest that the authors support their statements with research articles or systematic reviews/meta-analyses throughout the manuscript.

•Material and Methods

- Please provide more information about the study population in the text. There is no information regarding the number of subjects per group, ages, sexual orientation, etc. This information are only partially provided in tables. It appears that references #27 is a study previously performed by the group using the same sample population. I looked at the study, but the demographic information provided there is also incomplete. Please elaborate in detail on the characteristics of the study population.

- In study #27, the authors mention that the study population comprises young adults between 20-30 yrs of age. This information is not provided clearly in the present manuscript, however I assume that is true. There is a large discrepancy in the age of the subjects and the raters. It is well documented that age has a significant effect on the perception of attractiveness, with older individuals becoming “less strict” with time.

- The authors report keeping the first 16 PDs for their shape analysis (explaining more than 83% of variation). How did they decide on this cut off limit? Was the broken-stick method used?

- There is no mention in the manuscript regarding the assessment of systematic and random error related to the study methodology. In my view this is an essential part of the study methodology when conducting studies that investigate the subjective outcomes such as facial attractiveness.

•Results

- It appears that a 10% type-1 error was accepted for all statistical analyses. Can the authors please elaborate on this decision?

Minor comments:

•Title

-The title is a little confusing. It implies that carbohydrate consumption was measured with facial attractiveness. I suggest rephrasing the title to: “Refined carbohydrate consumption and facial attractiveness.”

•Abstract:

- Please include more specific results in the abstract. For example, how much was facial attractiveness decreased?

•Ethic Statement:

- The authors state that no authors had access to identifying information of the study participants. Were the authors not part of the study protocol that was submitted to the ethics committee? Did none of the authors participate in data collection and analyses? Could the authors please provide an “author contribution statement” in the revised manuscript?

•Introduction:

- There is a large number of references used in the first paragraph (2-14) to show an association between carbohydrate consumption and medical disease. I would suggest to reduce the number of references to the most relevant ones that apply more to the topic of interest.

•Material and Methods

- Menstrual cycle has an effect on facial attractiveness in women. Facial attractiveness increases during ovulation, as rated by male observers. This is a confounding factor that was not taken into consideration. Please see: “Roberts, S. C. et al. Female facial attractiveness increases during the fertile phase of the menstrual cycle. Proceedings of the Royal Society of London. Series B: Biological Sciences 271, 1–3 (2004).’

Also, women’s perception of male facial attractiveness is influenced by their menstrual cycle which would also affect ratings in this study. Please see: Penton-Voak, I. S. et al. Menstrual cycle alters face preference. Nature 399, 741–742 (1999).

These factors need to be discussed and considered as serious confounders in the methodology of this investigation.

- Was raters’ sexual orientation taken into consideration in the regression models or in rater selection?

•Figures and Tables

- Figures 2 and 3: Is the P-value a typo or did the authors accept a 10% type-1 error?

- I suggest that the authors reduce the text in the result section and add tables and figures displaying their results more visually. This will help readership the understand and interpret the results easier.

6. PLOS authors have the option to publish the peer review history of their article (what does this mean?). If published, this will include your full peer review and any attached files.

Reviewer #1: **Yes: **Ahsan

Reviewer #2: No

---

## [Author Response · Author response to Decision Letter 0]

3 Nov 2023

Thank you for the comments. They have all been taken into consideration, and the revised manuscript is significantly improved. 

Reviewer #1: I read the whole manuscript entitled " Chronic and immediate refined carbohydrate consumption measured by glycemic load, and facial attractiveness" and I have few suggestions.

- It's very hard to state attractiveness in words but it would be great if authors can define in few words/lines about it. Also, the opposite genders always have some degree of attractiveness to each other too. So, a formal statement about it should also be included in definition.

A sentence was added page 3, lines 76-78: “In the field of evolutionary biology, attractiveness (or preference) refers to an individual's tendency to be drawn to specific traits or characteristics in potential mating or social exchange partners.” 

- Page#3 and Line#64: I suggest changing the word “repeated” with persistent.

This word has been changed as suggested. 

- Page#3 and Line#68-69: any reference to support this would be great

We have added a reference page 3, line 67.

Reviewer #2: Thank you for the opportunity to review this very interesting study.

This is an observational study assessing the association between refined carbohydrate consumption and facial attractiveness. The authors have conducted a thorough investigation, taking into account a large number of variables that could potentially be related to the main outcome. Although I find great interest in the study and its results, I have some concerns regarding the methodology and study design. I find that the authors have not controlled well all possible confounders and have conducted a large number of tests in order to explore various associations. Also, the sample population is very small in order to make reliable conclusions about the primary outcome if all possible confounders are taken into account. Therefore, I am unfortunately not able to recommend publication of this manuscript in its current form. My detailed comments are listed below.

Major comments:

•Introduction

- I suggest that the authors rephrase and soften their statements about the association between carbohydrate consumption and medical disease. Despite current evidence regarding these relationships, most of these diseases have multifaceted etiologies and are likely not attributed to a single cause. I am referring to the first paragraph of the introduction.

In the first paragraph of introduction, we have moderated the importance of the link between refined carbohydrate consumption and medical problems page 3, lines 58-62: “The mismatch between how human physiology has evolved and Western industrialized lifestyles is seen as a contributing factor to the current epidemic of numerous medical problems. For example, it has been shown that this massive dietary change was involved in obesity, insulin resistance, type II diabetes, cardiovascular diseases, Alzheimer’s disease, hypertension or myopia (2–5).”

- The authors are citing Reference #8 to support a lot of their statement in the introduction. Reference #8 is a non-systematic, topic review article with little scientific merit. I suggest that the authors support their statements with research articles or systematic reviews/meta-analyses throughout the manuscript.

We have added reference to research articles or reviews throughout the manuscript each time this reference is cited (it is now reference #12). We agree that this reference is not a systematic review or meta-analyses. However, Cordain et al 2003 were the first to propose a plausible synthesis about the potential hormonal consequences of refined carbohydrate consumption on health. 

•Material and Methods

- Please provide more information about the study population in the text. There is no information regarding the number of subjects per group, ages, sexual orientation, etc. This information are only partially provided in tables. It appears that references #27 is a study previously performed by the group using the same sample population. I looked at the study, but the demographic information provided there is also incomplete. Please elaborate in detail on the characteristics of the study population.

Characteristics of the population are now provided in the Materials and Methods section, page 6, lines 138-144, and in the new Table 1 (corresponding to the former Table 4, completed). The order of the tables has been changed and the first paragraph of the Results section has been deleted. The population information contained in this paragraph has been moved to the Materials and Methods section for the sake of consistency (lines 165-167, page 8; lines 210-213,page 10; lines 234-235, page 11).

- In study #27, the authors mention that the study population comprises young adults between 20-30 yrs of age. This information is not provided clearly in the present manuscript, however I assume that is true. There is a large discrepancy in the age of the subjects and the raters. It is well documented that age has a significant effect on the perception of attractiveness, with older individuals becoming “less strict” with time.

This information on age of the study population in now clearly provided in Materials and Methods section, line 138 page 6. Effect of rater’s age on attractiveness (and also level study) was tested and was non significant for both sex. This is now indicated in manuscript page 13, lines 285-297, in a new paragraph “Effects of rater characteristics on their attractiveness perception” and results have been added in a new table in Supporting Information, Table S4. 

- The authors report keeping the first 16 PDs for their shape analysis (explaining more than 83% of variation). How did they decide on this cut off limit? Was the broken-stick method used?

The procedure is now explained in the text page 9 line 189-190: “ An arbitrary cut-off of minimum 80% variance explained was applied to select the axis, thus the first 16 axes were retained (explaining 83.7% of variance) for further analyses”.

- There is no mention in the manuscript regarding the assessment of systematic and random error related to the study methodology. In my view this is an essential part of the study methodology when conducting studies that investigate the subjective outcomes such as facial attractiveness.

We understand the importance of assessing systematic and random errors in studies investigating subjective outcomes such as facial attractiveness. While these aspects were not explicitly addressed in the first version of the manuscript, we would like to highlight that we have taken measures to account for them in our methodology. To assess systematic errors, we meticulously designed our study by following established protocols and ensuring that experimental conditions were standardized as much as possible (for subjects and raters). For example, we used recruitment procedures to limit the risk of selection bias by defining clear inclusion and exclusion criteria for our study samples. In addition, all photographs and all assessments of facial attractiveness were conducted in a controlled environment with consistent lighting, background, and room temperature to reduce variability caused by external factors. Participants and raters were also kept blind to the study's objectives to minimize potential biases related to knowledge of the research goals. Additionally, we employed appropriate statistical methods to control for potential confounding factors (for raters and subjects). Regarding random errors, we used reliable and reproducible measurement techniques for the photographs and the assessment of facial attractiveness. The sample sizes of participants and raters were determined on the basis of power calculations to ensure that the study had sufficient statistical power to detect significant differences in attractiveness ratings and we increased the sample sizes to minimize the impact of random variability (subjects and raters). This is now included in the manuscript in materials and methods, section “Assessment of systematic and random errors.“ (lines 339-356 page 15). 

•Results

- It appears that a 10% type-1 error was accepted for all statistical analyses. Can the authors please elaborate on this decision?

Type-I error is 5% throughout. There were several mentions in the text of “marginally significant” results when the P-value was between 0.05 and 0.10. To avoid confusion, these situations are now mentioned as “marginally non-significant” (page 16, lines 377, 389; page 17, lines 392, 405, 411). See also below for the path-analysis section (Minor comments / Figures and Tables).

Minor comments:

•Title

-The title is a little confusing. It implies that carbohydrate consumption was measured with facial attractiveness. I suggest rephrasing the title to: “Refined carbohydrate consumption and facial attractiveness.”

The title was rephrased: “Chronic and immediate Refined carbohydrate consumption and facial attractiveness”

•Abstract

- Please include more specific results in the abstract. For example, how much was facial attractiveness decreased?

More specific results are now included in the abstract, lines 38-45.

•Ethic Statement

- The authors state that no authors had access to identifying information of the study participants. Were the authors not part of the study protocol that was submitted to the ethics committee? Did none of the authors participate in data collection and analyses? Could the authors please provide an “author contribution statement” in the revised manuscript?

Some authors did participate at all stages of the study. Data were analyzed anonymously and no authors had access to information that could identify individual participants during or after data collection. We gave each participant a unique ID and each facial sample was anonymized with a separate set of ID. The translation key between these two sets of IDs was stored in a separate file. This file was stored in a secured server and was not stored beside the collected data. 

We have had a section “Author contribution in the revised manuscript” page 25 line 591-594. “CB and MR planned and supervised study. AV, CB, LG and VD conducted different parts of study. AV and CB analyzed study. AV, CB and MR wrote the manuscript. All authors read and approved the final manuscript.”

•Introduction:

- There is a large number of references used in the first paragraph (2-14) to show an association between carbohydrate consumption and medical disease. I would suggest to reduce the number of references to the most relevant ones that apply more to the topic of interest.

This was done. We have reduced the number of references.

•Material and Methods

- Menstrual cycle has an effect on facial attractiveness in women. Facial attractiveness increases during ovulation, as rated by male observers. This is a confounding factor that was not taken into consideration. Please see: “Roberts, S. C. et al. Female facial attractiveness increases during the fertile phase of the menstrual cycle. Proceedings of the Royal Society of London. Series B: Biological Sciences 271, 1–3 (2004).’

Also, women’s perception of male facial attractiveness is influenced by their menstrual cycle which would also affect ratings in this study. Please see: Penton-Voak, I. S. et al. Menstrual cycle alters face preference. Nature 399, 741–742 (1999).

These factors need to be discussed and considered as serious confounders in the methodology of this investigation.

Facial attractiveness is indeed influenced by many variables and we have taken into account the main ones, such as age, facial morphology, hormonal contraception, etc. But of course there are many variables not considered in this study, including menstrual cycles. This is compensated by the sample size, with the hypothesis that the sample is randomized according to the variables not included. In other words, if the stage of the menstrual cycle of each women participant is randomized for the sampling day, then there is no bias of not considering this variable. This limitation is now stated in the manuscript page 24, lines 567-573 and hormonal contraception was added in the new Table 1.

- Was raters’ sexual orientation taken into consideration in the regression models or in rater selection?

In order to reduce cultural sample heterogeneity, we kept for analyses heterosexual individuals (participants and raters).

This is now clearly indicated: line 138, page 6 for participants; line 229, page 11 for masculinity/femininity raters; line 250, page 12 for attractiveness raters.

•Figures and Tables

- Figures 2 and 3: Is the P-value a typo or did the authors accept a 10% type-1 error?

Structural equation modeling was performed using (conservatively) the variables from the model studying perceived attractiveness displaying P < 0.1, as explained in the text (page 14, lines 322-323). But all results described in this study are based on type-I error of 5%. This is now clearly stated. The legends of Figures 2 and 3 have been modified to avoid confusion.

- I suggest that the authors reduce the text in the result section and add tables and figures displaying their results more visually. This will help readership the understand and interpret the results easier.

We have added a new figure (Figure 4) which is a forest plot with the odds ratio from the model studying perceived attractiveness.

---

## [Decision Letter · Decision Letter 1]

22 Jan 2024

PONE-D-23-07591R1Chronic and immediate refined carbohydrate consumption and facial attractivenessPLOS ONE

Dear Dr. Berticat,

Thank you for submitting your manuscript to PLOS ONE. After careful consideration, we feel that it has merit but does not fully meet PLOS ONE’s publication criteria as it currently stands. Therefore, we invite you to submit a revised version of the manuscript that addresses the points raised during the review process.

We look forward to receiving your revised manuscript.

Kind regards,

Shen Liu, Ph.D

Academic Editor

PLOS ONE

Journal Requirements:

Additional Editor Comments:

I have carefully read the manuscript “Chronic and immediate refined carbohydrate consumption and facial attractiveness.” Note that I became a reviewer at this round; and my knowledge of the evolution of the manuscript is limited to what is present in the “response to reviewers” section. I gather that the revisions sufficiently addressed the previous concerns. I found much to like in the manuscript and found the topic novel and interesting. However, I think some further revisions may be necessary. I list potential revisions below for the editor and authors’ consideration.

1. My biggest issue with the present article was that I was unable to find any report of power analyses and could not discern how sample size was determined. In the revision, the authors explain in response to a reviewer that “The sample sizes of participants and raters were determined on the basis of power calculations to ensure that the study had sufficient statistical power to detect significant differences in attractiveness ratings, based on a previous study (8) and were chosen to minimize the impact of random variability.” The same text is also in the manuscript now. I was unable to find any details of power analyses in reference #8, (which, to be sure, should be here: https://journals.sagepub.com/doi/full/10.1177/1474704920960440). If power analyses indeed exist, they need to be reported. If they do not exist, then sensitivity analyses should be provided retrospectively and their result would critically determine the evaluation of the article (my decision recommendation of "minor revision" would probably not hold any longer). The current sample size is not so large by some standards (e.g., https://www.sciencedirect.com/science/article/pii/S0092656613000858) and we need to have a clearer idea of the reliability of the present findings.

2. My position amidst the incessant debates about null hypothesis significance testing is that usage of phrases such as “marginally significant” are in error. If you subscribe to the Neyman-Pearson approach, there is no such thing. I would recommend removing this phrase and being consistent in the usage of alpha level to decide on significance and its absence. In addition, there are many p-values derived from the same dataset and no consideration of experimentwise Type I error. Thus, if one also extends the threshold for deciding on the presence of effects to the “marginal” area (e.g., .05-.1), then there is an even greater risk of some of these findings to represent Type 1 error.

3. I did not see any mention of sleep quality (chronic and in the night prior to data collection) but based on the literature, this is another important factor in the current setting and could be mentioned as something that future studies should aim to control.

4. I did not see any mention of make-up or facial accessories. Was there any request for participants to remove make-up or detachable accessories (piercing, etc.)? If not this would be a serious limitation and could also be confounded with lifestyle and thus the chronic diet. Did any participants have facial tattoos? At the least, I would expect the authors to handle this by performing the same kind of procedure they applied to facial hairiness.

5. Response options for the “geographical origin of the grandparents” item could be added. It is not clear whether this is asking for ethnicity directly or the researchers are inferring ethnicity from geographical position of where the grandparents were born or grew up in.

6. Line 191: The full term for “LDA” should be added (i.e., “linear discriminant analysis” I suppose).

7. The authors describe where participants were approached but not where they were tested. The latter should be added.

8. Where only p-values are provided, full statistics could be added, such as on lines 234-235 (correlation coefficients should be added there). Assuming those are p-values, why are they capitalized?

9. I think the Statistical Analyses section would be easier to read if there was an overview of what was done and why at the beginning (and/or at the beginning of each of its subsections). Why was a particular analysis needed could be made clearer. In addition, I think the subsections should be grouped under broader headings. Which subsections are preparatory or necessary checks and which ones contain the central tests would become clearer this way.

10. It is not clear to me why parental home ownership was assessed. I assume it is a proxy for the participant’s socioeconomic status as these are relatively young individuals. Whatever it is assumed to measure, how is it linked to facial attractiveness (or how is it relevant to the current study)? These could be clarified.

11. In the parenthesis that starts at the end of line 310, shouldn’t one of the three modalities be “B2 versus B1” (instead of two of them both being “B1 versus B2”)?

12. In the captions for Figures 2 and 3, instead of “women/men to explain their attractiveness,” one could simply write “women’s/men’s attractiveness.”

13. Line 398: Shouldn’t “total deviance” be “total variance?”

14. Line 462: “influence” should be “influences”. In the same line, as in others, I think it’s better to follow an earlier reviewer’s suggestion of using “persistent” or “chronic” instead of “repeated.”

15. Lines 464-465: Delete the last “s” in both of these phrases: “men prefers” and “women prefers”

16. Line 465: I disagree with the assertion that “women prefers more masculine faces.” For instance, even the review article cited in support of this mentions the many nuances (e.g., contextual moderators) that would qualify such a broad assertion as well as findings in the opposite direction. Thus, I think at least an indication that these caveats are acknowledged should be provided. This is important because it will make the authors’ explanation of some of their effects more tentative, which I think would be more balanced at this stage of our knowledge about the topic.

17. Line 478: I do not understand the usage of “escalated”. How do these factors escalate (i.e., increase in intensity) to anything? I am not a native speaker but this does not make sense to me.

18. Line 536: I think this heading is not grammatical because of the phrase “evolutionary triggered.” The second paragraph under this heading seems unrelated to the heading.

19. Line 547: If the present study aimed to replicate reference #8, why not mention that in the introduction?

20. Line 574: Was this analysis reported anywhere? If no intention to report, then maybe the authors could add a note that the details are available upon request. Better, it could be added to the supplementary materials.

I hope my suggestions are useful and that at least some of them can be implemented with the effect of improving the manuscript.

Reviewers' comments:

Reviewer's Responses to Questions

**Comments to the Author**

1. If the authors have adequately addressed your comments raised in a previous round of review and you feel that this manuscript is now acceptable for publication, you may indicate that here to bypass the “Comments to the Author” section, enter your conflict of interest statement in the “Confidential to Editor” section, and submit your "Accept" recommendation.

Reviewer #3: (No Response)

2. Is the manuscript technically sound, and do the data support the conclusions?

Reviewer #3: Yes

3. Has the statistical analysis been performed appropriately and rigorously? 

Reviewer #3: Yes

4. Have the authors made all data underlying the findings in their manuscript fully available?

Reviewer #3: Yes

5. Is the manuscript presented in an intelligible fashion and written in standard English?

Reviewer #3: Yes

6. Review Comments to the Author

Reviewer #3: I have carefully read the manuscript “Chronic and immediate refined carbohydrate consumption and facial attractiveness.” Note that I became a reviewer at this round; and my knowledge of the evolution of the manuscript is limited to what is present in the “response to reviewers” section. I gather that the revisions sufficiently addressed the previous concerns. I found much to like in the manuscript and found the topic novel and interesting. However, I think some further revisions may be necessary. I list potential revisions below for the editor and authors’ consideration.

1. My biggest issue with the present article was that I was unable to find any report of power analyses and could not discern how sample size was determined. In the revision, the authors explain in response to a reviewer that “The sample sizes of participants and raters were determined on the basis of power calculations to ensure that the study had sufficient statistical power to detect significant differences in attractiveness ratings, based on a previous study (8) and were chosen to minimize the impact of random variability.” The same text is also in the manuscript now. I was unable to find any details of power analyses in reference #8, (which, to be sure, should be here: https://journals.sagepub.com/doi/full/10.1177/1474704920960440). If power analyses indeed exist, they need to be reported. If they do not exist, then sensitivity analyses should be provided retrospectively and their result would critically determine the evaluation of the article (my decision recommendation of "minor revision" would probably not hold any longer). The current sample size is not so large by some standards (e.g., https://www.sciencedirect.com/science/article/pii/S0092656613000858) and we need to have a clearer idea of the reliability of the present findings.

2. My position amidst the incessant debates about null hypothesis significance testing is that usage of phrases such as “marginally significant” are in error. If you subscribe to the Neyman-Pearson approach, there is no such thing. I would recommend removing this phrase and being consistent in the usage of alpha level to decide on significance and its absence. In addition, there are many p-values derived from the same dataset and no consideration of experimentwise Type I error. Thus, if one also extends the threshold for deciding on the presence of effects to the “marginal” area (e.g., .05-.1), then there is an even greater risk of some of these findings to represent Type 1 error.

3. I did not see any mention of sleep quality (chronic and in the night prior to data collection) but based on the literature, this is another important factor in the current setting and could be mentioned as something that future studies should aim to control.

4. I did not see any mention of make-up or facial accessories. Was there any request for participants to remove make-up or detachable accessories (piercing, etc.)? If not this would be a serious limitation and could also be confounded with lifestyle and thus the chronic diet. Did any participants have facial tattoos? At the least, I would expect the authors to handle this by performing the same kind of procedure they applied to facial hairiness.

5. Response options for the “geographical origin of the grandparents” item could be added. It is not clear whether this is asking for ethnicity directly or the researchers are inferring ethnicity from geographical position of where the grandparents were born or grew up in.

6. Line 191: The full term for “LDA” should be added (i.e., “linear discriminant analysis” I suppose).

7. The authors describe where participants were approached but not where they were tested. The latter should be added.

8. Where only p-values are provided, full statistics could be added, such as on lines 234-235 (correlation coefficients should be added there). Assuming those are p-values, why are they capitalized?

9. I think the Statistical Analyses section would be easier to read if there was an overview of what was done and why at the beginning (and/or at the beginning of each of its subsections). Why was a particular analysis needed could be made clearer. In addition, I think the subsections should be grouped under broader headings. Which subsections are preparatory or necessary checks and which ones contain the central tests would become clearer this way.

10. It is not clear to me why parental home ownership was assessed. I assume it is a proxy for the participant’s socioeconomic status as these are relatively young individuals. Whatever it is assumed to measure, how is it linked to facial attractiveness (or how is it relevant to the current study)? These could be clarified.

11. In the parenthesis that starts at the end of line 310, shouldn’t one of the three modalities be “B2 versus B1” (instead of two of them both being “B1 versus B2”)?

12. In the captions for Figures 2 and 3, instead of “women/men to explain their attractiveness,” one could simply write “women’s/men’s attractiveness.”

13. Line 398: Shouldn’t “total deviance” be “total variance?”

14. Line 462: “influence” should be “influences”. In the same line, as in others, I think it’s better to follow an earlier reviewer’s suggestion of using “persistent” or “chronic” instead of “repeated.”

15. Lines 464-465: Delete the last “s” in both of these phrases: “men prefers” and “women prefers”

16. Line 465: I disagree with the assertion that “women prefers more masculine faces.” For instance, even the review article cited in support of this mentions the many nuances (e.g., contextual moderators) that would qualify such a broad assertion as well as findings in the opposite direction. Thus, I think at least an indication that these caveats are acknowledged should be provided. This is important because it will make the authors’ explanation of some of their effects more tentative, which I think would be more balanced at this stage of our knowledge about the topic.

17. Line 478: I do not understand the usage of “escalated”. How do these factors escalate (i.e., increase in intensity) to anything? I am not a native speaker but this does not make sense to me.

18. Line 536: I think this heading is not grammatical because of the phrase “evolutionary triggered.” The second paragraph under this heading seems unrelated to the heading.

19. Line 547: If the present study aimed to replicate reference #8, why not mention that in the introduction?

20. Line 574: Was this analysis reported anywhere? If no intention to report, then maybe the authors could add a note that the details are available upon request. Better, it could be added to the supplementary materials.

I hope my suggestions are useful and that at least some of them can be implemented with the effect of improving the manuscript.

7. PLOS authors have the option to publish the peer review history of their article (what does this mean?). If published, this will include your full peer review and any attached files.

Reviewer #3: **Yes: **S. Adil Saribay

---

## [Author Response · Author response to Decision Letter 1]

1 Feb 2024

Thank you for the comments. They have all been taken into consideration, and the revised manuscript is significantly improved. 

Reviewer #3: I have carefully read the manuscript “Chronic and immediate refined carbohydrate consumption and facial attractiveness.” Note that I became a reviewer at this round; and my knowledge of the evolution of the manuscript is limited to what is present in the “response to reviewers” section. I gather that the revisions sufficiently addressed the previous concerns. I found much to like in the manuscript and found the topic novel and interesting. However, I think some further revisions may be necessary. I list potential revisions below for the editor and authors’consideration.

1. My biggest issue with the present article was that I was unable to find any report of power analyses and could not discern how sample size was determined. In the revision, the authors explain in response to a reviewer that “The sample sizes of participants and raters were determined on the basis of power calculations to ensure that the study had sufficient statistical power to detect significant differences in attractiveness ratings, based on a previous study (8) and were chosen to minimize the impact of random variability.” The same text is also in the manuscript now. I was unable to find any details of power analyses in reference #8, (which, to be sure, should be here: https://journals.sagepub.com/doi/full/10.1177/1474704920960440). If power analyses indeed exist, they need to be reported. If they do not exist, then sensitivity analyses should be provided retrospectively and their result would critically determine the evaluation of the article (my decision recommendation of "minor revision" would probably not hold any longer). The current sample size is not so large by some standards (e.g., https://www.sciencedirect.com/science/article/pii/S0092656613000858) and we need to have a clearer idea of the reliability of the present findings.

The explanation of the choice of sample size based on our pilot study has been given in lines 350 to 352, page 15, and the sentence has been reworded. We agree that our sample size is not very large. This is why we now present the details of the sensibility analysis (randomly deleting 10% of the data and computing p-values, then repeating this process 1000 times, and analyzing the resulting p-value distributions for each variable) in the new Table S6. 

2. My position amidst the incessant debates about null hypothesis significance testing is that usage of phrases such as “marginally significant” are in error. If you subscribe to the Neyman-Pearson approach, there is no such thing. I would recommend removing this phrase and being consistent in the usage of alpha level to decide on significance and its absence. In addition, there are many p-values derived from the same dataset and no consideration of experiment wise Type I error. Thus, if one also extends the threshold for deciding on the presence of effects to the “marginal” area (e.g., .05-.1), then there is an even greater risk of some of these findings to represent Type 1 error.

We agree. The only occurrence of “marginally significant” was replaced with “marginally non-significant” (line 501 page 22). Thus, the threshold of significance is maintained at 0.05 throughout. Thus type-I error are not inflated due to a variable threshold. 

3. I did not see any mention of sleep quality (chronic and in the night prior to data collection) but based on the literature, this is another important factor in the current setting and could be mentioned as something that future studies should aim to control.

We now mention this point in the limitations section and have added a reference.

4. I did not see any mention of make-up or facial accessories. Was there any request for participants to remove make-up or detachable accessories (piercing, etc.)? If not this would be a serious limitation and could also be confounded with lifestyle and thus the chronic diet. Did any participants have facial tattoos? At the least, I would expect the authors to handle this by performing the same kind of procedure they applied to facial hairiness.

We indeed asked participant to remove their make-up and all detachable accessories. In the manuscript, we already stated on page 6, line 135 to 136 “The subjects were asked to express a neutral face (without a smile), to tie their hair and to remove any glasses or earrings”. We added clarifications as suggested “The subjects were asked to express a neutral face (without a smile), to tie their hair and to remove any glasses, earrings, piercing or make-up.” No participant was wearing a facial tattoo. Indeed, absence of facial tattoo was a condition for participation, and this is now stated (line 115, page 5). 

5. Response options for the “geographical origin of the grandparents” item could be added. It is not clear whether this is asking for ethnicity directly or the researchers are inferring ethnicity from geographical position of where the grandparents were born or grew up in.

We asked participants to specify the continent each of their grandparents came from in order to reduce cultural heterogeneity (only participants with their 4 grandparents of European origin were selected). The text has been modified to specify this aspect (line 120-121 page 5). 

6. Line 191: The full term for “LDA” should be added (i.e., “linear discriminant analysis” I suppose).

This was done. We added the full term.

7. The authors describe where participants were approached but not where they were tested. The latter should be added.

We added this page 5 lines 116 and replaced the sentence “Subjects were given an appointment early in the morning and had to come in on an empty stomach in groups of three to four” with “Subjects were given an early-morning appointment and were asked to come to our laboratory for the experiments in groups of three or four on an empty stomach”.

8. Where only p-values are provided, full statistics could be added, such as on lines 234-235 (correlation coefficients should be added there). Assuming those are p-values, why are they capitalized?

We have replaced “P” with “p” for p-values throughout the manuscript. We have added the full statistics when p-values were provided without a table.

9. I think the Statistical Analyses section would be easier to read if there was an overview of what was done and why at the beginning (and/or at the beginning of each of its subsections). Why was a particular analysis needed could be made clearer. In addition, I think the subsections should be grouped under broader headings. Which subsections are preparatory or necessary checks and which ones contain the central tests would become clearer this way.

An overview of what was done and why has been added lines 261 to 264 page 12. A more general heading has also been added above the subsections devoting to raters’ characteristics.

10. It is not clear to me why parental home ownership was assessed. I assume it is a proxy for the participant’s socioeconomic status as these are relatively young individuals. Whatever it is assumed to measure, how is it linked to facial attractiveness (or how is it relevant to the current study)? These could be clarified.

Indeed, parental home ownership was assessed as a proxy of the socioeconomic status of the young participants. Higher socioeconomic status may provide individuals with better access to resources, including healthcare, nutrition or education. This can contribute to overall health and well-being, potentially influencing facial features associated with attractiveness. Moreover, individuals may be attracted to traits associated with higher socioeconomic status due to perceptions of stability, resources and overall fitness. Socioeconomic status is therefore potentially linked to attractiveness and diet. This is now precised line 122 page 5 and a reference has been added page 14 line 313. 

11. In the parenthesis that starts at the end of line 310, shouldn’t one of the three modalities be “B2 versus B1” (instead of two of them both being “B1 versus B2”)?

Thank you, it was a mistake. This is now corrected.

12. In the captions for Figures 2 and 3, instead of “women/men to explain their attractiveness,” one could simply write “women’s/men’s attractiveness.”

This has been modified.

13. Line 398: Shouldn’t “total deviance” be “total variance?”

As we have used logistic regression, we must thus speak of deviance rather than variance.

14. Line 462: “influence” should be “influences”. In the same line, as in others, I think it’s better to follow an earlier reviewer’s suggestion of using “persistent” or “chronic” instead of “repeated.”

These points have now been corrected.

15. Lines 464-465: Delete the last “s” in both of these phrases: “men prefers” and “women prefers”

This has been corrected.

16. Line 465: I disagree with the assertion that “women prefers more masculine faces.” For instance, even the review article cited in support of this mentions the many nuances (e.g., contextual moderators) that would qualify such a broad assertion as well as findings in the opposite direction. Thus, I think at least an indication that these caveats are acknowledged should be provided. This is important because it will make the authors’ explanation of some of their effects more tentative, which I think would be more balanced at this stage of our knowledge about the topic.

We agree and this was corrected in the sentence lines 460 to 462 page 20.

17. Line 478: I do not understand the usage of “escalated”. How do these factors escalate (i.e., increase in intensity) to anything? I am not a native speaker but this does not make sense to me.

We have replaced “escalated” by “lead to”.

18. Line 536: I think this heading is not grammatical because of the phrase “evolutionary triggered.” The second paragraph under this heading seems unrelated to the heading.

The second paragraph has been deleted and integrated into the Introduction (see response to the following question).

19. Line 547: If the present study aimed to replicate reference #8, why not mention that in the introduction?

This is now clearly indicated in Introduction, lines 82, 98 to 99, 101 to 103 page 4. This is the paragraph deleted in question 18 which has been incorporated here.

20. Line 574: Was this analysis reported anywhere? If no intention to report, then maybe the authors could add a note that the details are available upon request. Better, it could be added to the supplementary materials.

This analysis is now presented in the new Table S6.

---

## [Editor Report · Decision Letter 2]

2 Feb 2024

Chronic and immediate refined carbohydrate consumption and facial attractiveness

PONE-D-23-07591R2

Dear Dr. Berticat,

We’re pleased to inform you that your manuscript has been judged scientifically suitable for publication and will be formally accepted for publication once it meets all outstanding technical requirements.

Kind regards,

Shen Liu, Ph.D

Academic Editor

PLOS ONE
---

## [Editor Report · Acceptance letter]

15 Feb 2024

PONE-D-23-07591R2 

PLOS ONE

Dear Dr. Berticat, 

I'm pleased to inform you that your manuscript has been deemed suitable for publication in PLOS ONE. Congratulations! Your manuscript is now being handed over to our production team.

Kind regards, 

on behalf of

Prof. Shen Liu 

Academic Editor

PLOS ONE